# Minerochemical and Microtextural Study of the Ungrouped Iron Meteorite Oglat Sidi Ali, Eastern Highlands, Morocco, and Geomorphological Characterization of Its Strewnfield

Hassane Nachit [1], Abderrahmane Ibhi [1], Mohamed En-nasiry [1], Vanni Moggi Cecchi [2], Giovanni Pratesi [3,4], Christopher D. K. Herd [5] and Giorgio S. Senesi [6,*]

1  Petrology, Metallogeny and Meteorites Team, Faculty of Sciences, Ibn Zohr University, 80000 Agadir, Morocco
2  Museo di Storia Naturale—Sistema Museale di Ateneo, Università degli Studi di Firenze, 50121 Firenze, Italy
3  Dipartimento di Scienze della Terra, Università degli Studi di Firenze, 50121 Firenze, Italy
4  INAF-Istituto di Astrofisica e Planetologia Spaziali, 00133 Roma, Italy
5  Department of Earth and Atmospheric Sciences, University of Alberta, Edmonton, AB T6G 2E3, Canada
6  CNR, Istituto per la Scienza e Tecnologia dei Plasmi (ISTP), sede di Bari, 70126 Bari, Italy
*  Correspondence: giorgio.senesi@cnr.it; Tel.: +390805929505

**Abstract:** Fragments of a new iron meteorite were found in and collected from Oglat Sidi Ali, Maatarka region, Morocco, during a series of expeditions in the years 2013–2017. The physical characteristics of recovered fragments feature typical attributes of individual samples of a unique meteorite strewnfield that originated from an iron meteorite shower via the fragmentation of a single body that broke up in the lower atmosphere. The total recovered mass of the Oglat Sidi Ali meteorite fragments was estimated to amount to more than 800 kg spread across a NE–SW oriented, 20 km-long and 5 km-wide strewnfield. Geochemical and mineralogical data achieved on Oglat Sidi Ali fragments, as well as the analysis of its microstructures obtained using electron backscattered diffraction (EBSD), suggested it should be classified as an ungrouped iron meteorite. A comparison of this meteorite with other ungrouped iron meteorites, such as NWA 859 and NWA 11010, purchased between 2001 and 2016 in various cities of Northeast Morocco show apparently similar mineralogy, geochemistry and textural features, suggesting a common origin from a single extraterrestrial body.

**Keywords:** Oglat Sidi Ali; iron meteorite; mineralogy; geochemistry; strewnfield; electron backscattered diffraction

## 1. Introduction

There is no doubt that desert meteorites play an increasingly important role in the study of cosmochemistry due to their relatively low recovery costs and generally unrestricted collection and flow in most parts of the world. Several countries have collected a large number of meteorites in the Antarctic. However, most of the Antarctic meteorites have changed their positions after they landed on the Earth's surface, and thus, it is almost impossible to obtain information such as the entry angle, entry direction and breakup of meteoroids. In most cases, the locations of desert meteorites represent the landing sites, which provide important information for studying the mechanism of meteoroid entry and break up. Meanwhile, the location information is significantly important to judge whether the meteorite fragments are pairing or independent falls.

Morocco is an excellent place for finding meteorites thanks to the large extensions of prospective areas in the Sahara, which extend thousands of square kilometers without any obstacle. These areas are very dry with a semi-arid climate and very low erosion, and thus, meteorites can be preserved for a long time [1,2]. Currently, 41 different iron meteorites from Morocco are classified and officially approved by the Nomenclature Committee of The Meteoritical Society [3].

Since 1998, an extensive and long-lasting meteorite search has been conducted by nomads and private and public prospectors in the Oglat Sidi Ali strewnfield, which led to the discovery and collection of a large number of meteorite fragments. Field data would suggest that the recovered samples are part of a unique strewnfield produced by a single body that broke up in the lower atmosphere (Figure 1 left). Analytical data performed on an etched and polished section suggested a classification of an ungrouped iron meteorite. Some of these data were used to classify the meteorite fragments for their submission to, and official approval by, the Nomenclature Committee of the Meteoritical Society [4].

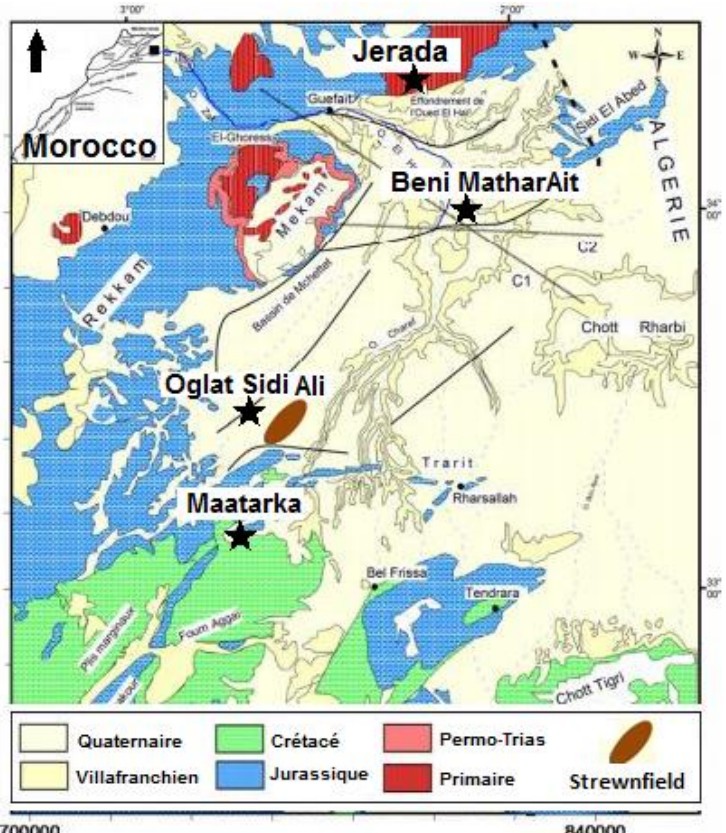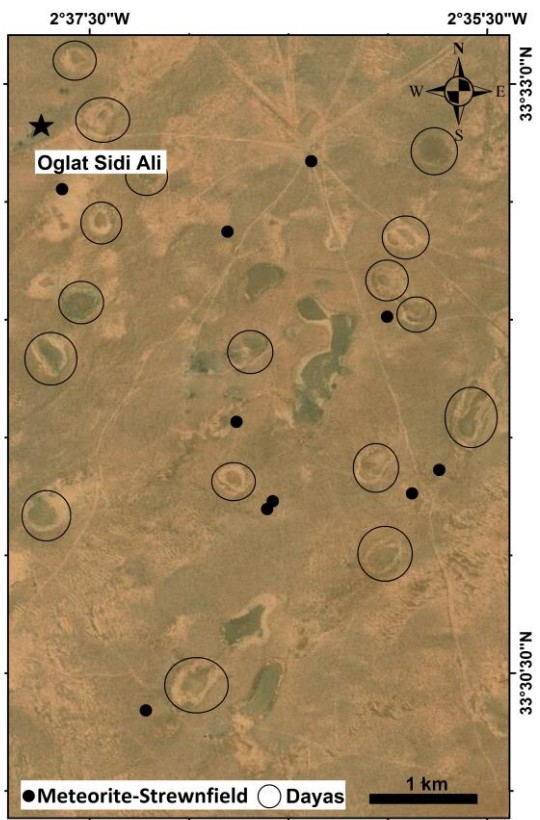

**Figure 1.** The strewnfield in the Eastern Highlands of Morocco (**left**). Circular structures called Dayas and the dispersed field of meteorite fragments (**right**).

The objectives of this work were to (a) investigate the overall geomorphology of the Oglat Sidi Ali meteorite scattering area, measure the size of the meteorite strewnfield and infer the orientation of the meteoroid trajectory; (b) determine the distribution, sizes, shapes and masses of meteorite fragments; (c) identify the nature of the meteorite fragments by measuring microtextural features, mineralogy and chemical characteristics; (d) compare the textural description, electron backscattered diffraction (EBSD) and trace element data of this meteorite to those of a set of three ungrouped iron meteorites from North Morocco; and (e) discuss the possible genetic relationship between the Oglat Sidi Ali meteorite and other Moroccan ungrouped plessitic meteorites.

## 2. Geomorphology of the Oglat Sidi Ali Site

The Eastern Highlands of Morocco consist of an essentially Jurassic-Cretaceous block of the crust bounded by the Middle and High Atlas Mountains. This vast plateau is barely tectonized and slightly tilted to the south without significant alpine deformation [5,6]. The tabular structure of this area is indented by Devonian and Carboniferous buttonholes (Debdou, Mekkam, Aouizert and Lalla Mimouna), and is displaced by faults that show a general NE–SW direction [7]. Under the Neogene sediment, the Atlas substrata seem

to be dissociated in horst and grabens in the form of large sub-tabular or slightly wavy panels [8].

The Oglat Sidi Ali site is located in the Eastern Highlands of Morocco, about 20 km north of the Maatarka region (coordinates 33°31′45″ N, 02°37′01″ W) (Figure 1, left). The scattering area is crossed by the N19 road to the south and west, as well as the roads N17 and R606 to the east and north, respectively. The site consists of a desert flat area covered by Quaternary sediments, which are mainly conglomerates featuring a few older rocks exposed on the surface. Many enigmatic circular structures called "Dayas" can be identified via visual inspection and aerial photography in the field of dispersed meteorite fragments (Figure 1 right). The largest structure, named Dayas Lahmar, is about 400 m in diameter and 5 to 6 m deep. No specific research has been conducted so far on these rounded depressions, and their origin and date of formation remain unknown. However, the study of conglomerates (Figure 2, left) located around these features indicates that they consist of poorly sorted lithoclasts of limestone, quartz and opaque minerals of size ranging from a few millimeters to a few centimeters, which are cemented in a carbonate or ferruginous matrix.

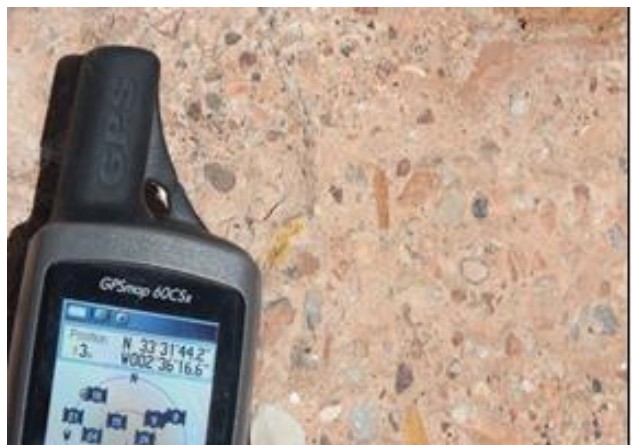 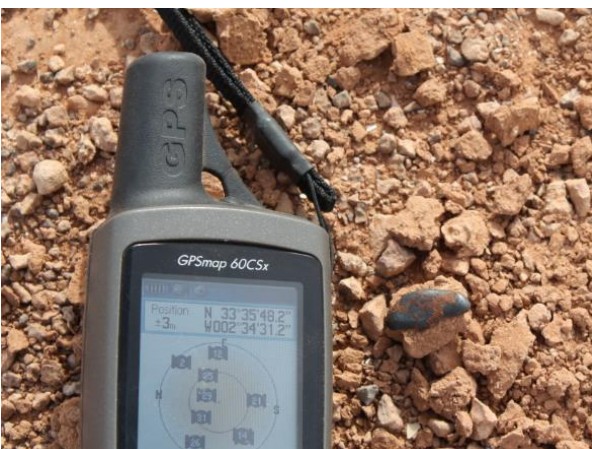

**Figure 2.** Conglomerates of the Oglat Sidi Ali site (**left**); a 30 g meteorite sample, buried at a depth of a few centimeters (**right**).

Erosion occurs widely in this area with the presence of small ravines resulting from rare and violent rains, deflation and corrosion by winds that attack the bare ground. Due to the flat topography and climate conditions, meteorite fragments are clearly visible on the surface or buried at a depth of a few centimeters (Figure 2 right). The fragments are probably still in their initial position or experienced only very short transport.

## 3. The Oglat Sidi Ali Strewnfield

Since 1998, several thousand small- (<10 kg), medium- (10 to 50 kg) and large-sized (>50 kg) meteorite fragments have been recovered in the Oglat Sidi Ali area, with more than 800 kg collected over 18 years, although some sources speculate that a larger amount has been recovered (up to 1 ton). We summarize some of the main recoveries in Table 1.

**Table 1.** Some information on the fragments of meteorite found in the region of Oglat Sidi Ali.

| Year | Meteorite Characterizations | GPS Coordinates |
|------|------------------------------|------------------|
| 1998 | Three specimens weighing about 1 kg were collected in this area by a nomad who sold them to a dealer in Oujda. | 33°29′46.5″ N, 02°36′27.6″ W<br>33°32′44.1″ N, 02°36′22.4″ W<br>33°31′20.6″ N, 02°36′26.1″ W |

**Table 1.** *Cont.*

| Year | Meteorite Characterizations | GPS Coordinates |
|---|---|---|
| 2001–2005 | The meteorite hunters conducted a systematic search with the aid of metal detectors: <br><br> - Hundreds of samples of 1–10 g, many of 100–1000 g and a few samples >1 kg were recovered. <br> - Three large samples, weighing 15, 25 and 72 kg, respectively, were recovered from 5 to 20 cm below the surface. | 33°36′5.7″ N, 02°34′54.6″ W <br> 33°31′15.3″ N, 02°34′33.4″ W <br> 33°33′12.9″ N, 02°34′58.8″ W <br> 33°30′16.8″ N, 02°36′47.3″ W <br> 33°31′21.4″ N, 02°36′25.3″ W <br> 33°30′14.0″ N, 02°36′39.8″ W <br> 33°26′55.7″ N, 02°39′36.1″ W |
| 2012 | A single mass of approximately 139 kg was recovered from a depth of about 60 cm. | 33°28′01.0″ N, 02°39′17.2″ W |
| 2013, 2014, 2015 and 2017 | Four fieldwork expeditions were conducted by some of the authors with the aid of a metal detector and the collaboration of dozens of nomads and dealers to get information about old finds, collect new fragments (Figures 3 and 4), and define the size of the strewnfield and the direction of the fall. | 33°31′33.1″ N, 02°36′24.8″ W <br> 33°31′39.2″ N, 02°36′34.8″ W <br> 33°31′54.1″ N, 02°36′39.2″ W <br> 33°32′12.0″ N, 02°36′33.8″ W <br> 33°31′59.2″ N, 02°35′27.9″ W <br> 33°33′29″ N, 02°35′59.2″ W <br> 33°29′58.7″ N, 02°36′25.1″ W |

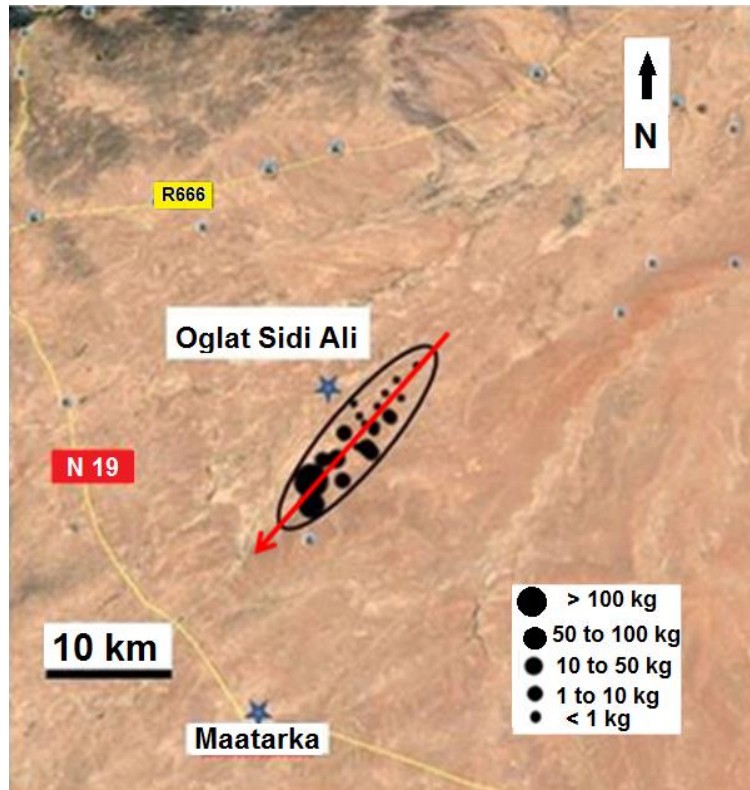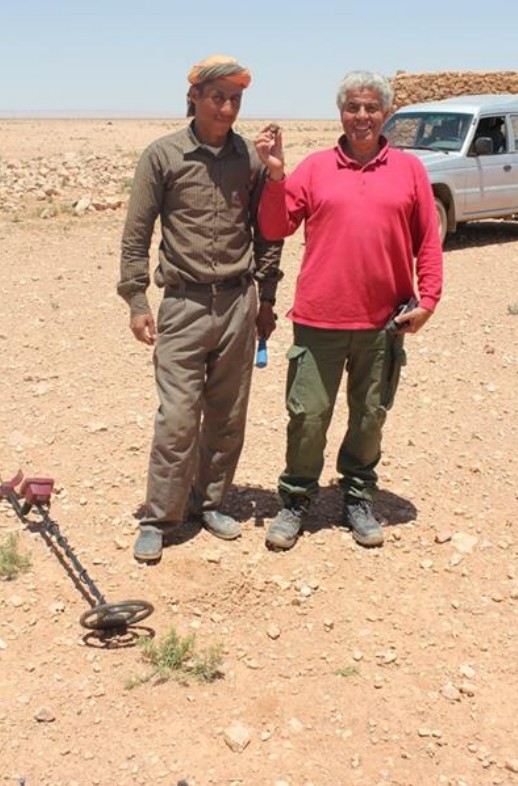

**Figure 3.** Strewnfield of the Oglat Sidi Ali meteorite (**left**). Prof. Nachit with a nomad of the region with a meteorite fragment find (**right**).

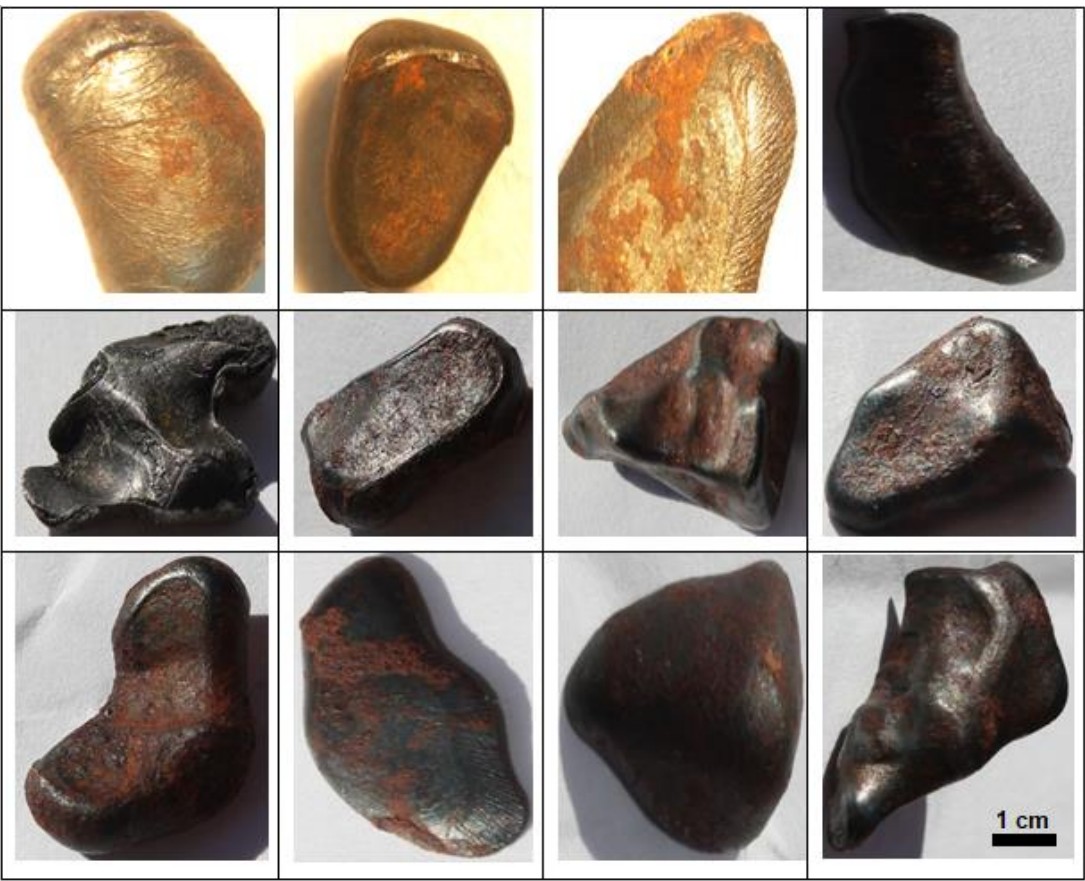

**Figure 4.** Physical characteristics of individual samples of the Oglat Sidi Ali iron meteorite.

The dispersion of the recovered specimens, as well as their smooth or regmaglypted appearance (see Section 5), indicates that they are "individual" samples that originated from a single fragmentation event of a body in the lower atmosphere. The meteorite shower caused by the explosion occurred over a large oval-shaped scattering area of approximately 70 km$^2$ (Figure 3 left). Analysis of the fragment size pattern over the strewnfield allowed us to infer the original flight direction of the meteoroid. In general, the smaller fragments tend to fall earlier, whereas the biggest fragments fall later, with the main mass of the meteorite falling in the downrange part of the scattering ellipse [9]. In the case of the Oglat Sidi Ali strewnfield, thousands of fragments were scattered in an area about 20 km long and 5 km wide in the NE–SW direction with smaller specimens (1 g to 1 kg) toward the NE and larger ones (72 to 139 kg) toward the SW of the strewnfield, which suggests that the flight path of the meteoroid was from NE to SW (Figure 3).

As the target field of Oglat Sidi Ali has a flat surface, the distribution of meteorite fragments depends on the entry angle, preatmospheric mass and strength of the meteoroid. An entry angle of 90° produces a circular scattering area, whereas lower entry angles determine elongated elliptical strewnfields. Computer simulations of strewnfield formation showed that for entry angles of 45°, similar to that of Oglat Sidi Ali, the scattering area length was greater than 11 km [10]. Furthermore, as the number of small individual fragments was demonstrated to increase markedly with decreasing meteoroid speed [10], the large amount of relatively small fragments in the strewnfield would indicate a low speed of the meteoroid.

## 4. Materials and Methods

During three expeditions (2013, 2014 and 2015) of exploration fieldwork, we recovered 72 specimens for a total weight of approx. 600 g. From these specimens, 42 pieces, totaling 195 g, are on deposit at the Ibn Zohr University, Morocco. The type specimen, consisting

of three small pieces weighing 30 g in total, is on deposit at the Museo di Storia Naturale dell'Università di Firenze, Italy (inventory # MSN-Fi I3665), while two more pieces, totaling 14.5 g and including an etched and polished mount, are deposited at the Museo di Scienze Planetarie of Prato, Italy. Oglat Sidi Ali was submitted to the Nomenclature Committee of the Meteoritical Society and approved as an ungrouped iron meteorite in 2015 [4].

A first-hand textural investigation was performed with a Zeiss metallographic optical microscope at the Laboratories of the Ibn-Zohr University, whereas more detailed textural features were determined by means of the electron backscattered diffraction (EBSD) technique at the Centro di Servizi di Microscopia Elettronica e Microanalisi (MEMA) laboratories of the Università di Firenze using a ZEISS EVO MA15 instrument equipped with an Oxford Symmetry S2 EBSD detector. Meteorite samples were prepared using standard metallographic polishing procedures followed by a two-step vibratory polishing procedure. The first vibratory polish was performed using 0.1 μm $Al_2O_3$ polishing paste for 4 h, followed by a second vibratory polishing step using 0.04 μm $SiO_2$ for 2–4 h. Both vibratory polishing steps were carried out on a napped cloth. This procedure removed the surface deformations caused by standard metallographic polishing. Diffraction patterns were obtained using a beam voltage of 20 kV, a beam current of 1–2 nA and a spot size diameter of <2 nm. The orientation data were acquired pixel-by-pixel with a pixel spacing of 0.02–0.5 μm.

Scanning electron microscope (SEM) images, energy dispersive X-ray (EDX) analyses and compositional X-ray maps were performed at the MEMA Laboratories of the Università di Firenze by means of a ZEISS EVO40 and an EVO MA15 instruments. Electron microprobe analyzer–wavelength dispersive spectrometers (EMPA-WDS) analyses were performed at the LaMA laboratory of the Dipartimento di Scienze della Terra of the Università di Firenze by means of a JEOL-JXA 8230 electron microprobe equipped with 5 WDS spectrometers and an EDS solid state detector (SDD). The following Astimex standards were used for the EMPA-WDS analyses: metallic Fe, Ni, Co, Ge and Cu for Fe, Co, Ni, Ge and Cu in Fe-Ni alloys; marcasite for Fe and S, metallic Ni and Co for Ni and Co, chromite for Cr and apatite for P in troilite; and metallic Fe, Ni, Co and Si for Fe, Ni, Co, Si and apatite for P in nickel-phosphide.

Minor and trace element analyses were performed at the Department of Earth and Atmospheric Sciences, University of Alberta, Canada, on an approximately 400 mg interior fragment of the meteorite. The sample was dissolved in at least 5 mL of 8 N $HNO_3$ acid, with dissolution taking place in Teflon containers. Concentrations of trace elements were determined using a Thermo Scientific™ iCAP™ RQ inductively coupled plasma mass spectrometer (ICP-MS). Elements were determined using the following isotopes: $^{59}$Co, $^{60}$Ni and $^{61}$Ni (average of the two), $^{63}$Cu and $^{65}$Cu (average of the two), $^{69}$Ga, $^{75}$As, $^{101}$Ru and $^{102}$Ru (average of the two), $^{105}$Pd, $^{182}$W and $^{184}$W (average of the two), $^{185}$Re, $^{189}$Os and $^{190}$Os (average of the two), $^{191}$Ir and $^{193}$Ir (average of the two), $^{194}$Pt and $^{195}$Pt (average of the two), and $^{197}$Au. A sample from North Chile (Filomena) of similar mass was carried through the preparation and analysis procedure and used as a secondary standard. Although Cr and Ge were sought, they could not be reliably determined using this method. Note that our method improves upon the method used to obtain data for Meteoritical Bulletin submission [4].

The measurement of Ge was performed at the Istituto Nazionale di Fisica (INFN) laboratories in Firenze with the KN3000 Van de Graaff accelerator using the external-beam particle-induced X-ray emission (PIXE) setup described by Fedi et al. [11]. The 3 MeV proton beam passes through a 140 μm graphite collimator (length of tenths of millimeters) and is extracted through a Kapton exit window that is 7.5 μm thick. The beam diameter impinging on the sample, after traveling about 1 cm in the atmosphere, is about 500 μm. In order to change the beam energy on the sample and reduce the pile-up effect due to major elements (Fe and Ni), five energy degraders, each one made of a 50 μm Al foil, were interposed between the exit window and the sample. The absorption ratio produced by the Al foils is 0.38.

## 5. Results and Discussion

### 5.1. Physical Characteristics

Most recovered specimens and those observed by dealers exhibited various shapes and feature smooth surfaces covered by cavities (regmaglypts) that were formed by ablation due to the frictional heating that occurred during the atmospheric passage of the meteorite (Figure 4). However, some individuals were rounded and presented only traces of fused pockets (sample sizes between 5 and 6 cm). Although the surfaces of many specimens were reddish with a minor degree of oxidation, oxidation did not penetrate deeply into the interiors of the fragments, which suggested a relatively short residence time on the terrestrial soil and a very low weathering grade. Many pieces were covered by flight-oriented flow lines and the remains of a blue-black fusion crust, which probably originated from the melting of the surface due to air friction.

### 5.2. Textural Features

Although the surfaces of many specimens were reddish, the cut surfaces of the interiors of most samples were fresh and displayed no traces of staining, which suggested a very low weathering grade. Low-magnification optical microscope analyses performed with a binocular stereomicroscope on a cut-and-etched surface allowed for the examination of the internal texture. The meteorite displayed a "plessitic octahedrite" texture consisting of elongated spindles of kamacite set in a groundmass of a fine-grained plessitic intergrowth of kamacite and taenite.

The etched surface showed exsolution lamellae with four sets of bands with different angles of intersection, which displayed a distinctive Widmanstätten pattern of long bundled kamacite lamellae with pointed ends. The kamacite formed slender spindles, which are sharp, discontinuous and bordered by taenite. The lamellae were frequently grouped and had an average bandwidth of 0.3 ± 0.1 mm, which indicated a very fine octahedrite structure.

Backscattered electron (BSE) images obtained using SEM revealed a much finer scale pattern of these lamellae that consisted of multiple tiny kamacite spindles roughly parallel to each other and ranging in width from 30 to 80 μm, which were separated by thin, Ni-rich taenitic spindles, both forming a plessitic octahedrite arrangement (Figure 5b). Small troilite and scattered nickel-phosphide grains were also observed.

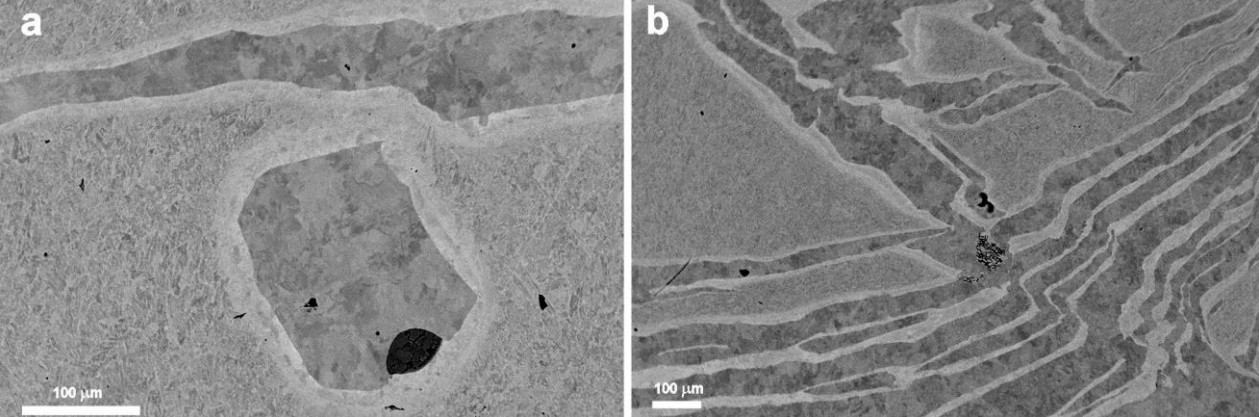

**Figure 5.** SEM-BSE image of two areas of a polished sample of the Oglat Sidi Ali meteorite; dark grey is kamacite; pale grey is taenite and Ni-rich taenite; medium grey is a plessitic texture; black areas are troilite (**a**) and nickel-phosphide (**b**).

More detailed information was obtained by means of EBSD that allowed for distinguishing the body-centered cubic (bcc) alpha phase (kamacite) from the face-centered cubic (fcc) gamma phase (taenite) of Fe-Ni alloys thanks to their electron diffraction patterns, as well as determining the orientation of the bcc and fcc phases relative to the sample

placement (x, y and z directions) under the electron beam. In particular, the orientation maps, or inverse pole figure maps, of the scanned specimen surface were developed using a color scheme to represent the orientation of the bcc and fcc phases relative to the major poles (111, 100, 110) of the stereographic triangle (Figure 6) so that pixels with the same orientation had the same color. However, the sample regions shown in black could not be indexed as fcc or bcc Fe-Ni because of their small dimensions. Rogue or misindexed pixels were removed and then filled using a routine eight-nearest-neighbor hole-filling that compared the surrounding pixels and fills in the missing ones. The orientation maps were used to determine both the grain size of the different portions of the selected areas and the local crystallographic orientations, which provided critical information that allowed for understanding how the various phases formed during reheating and the subsequent cooling.

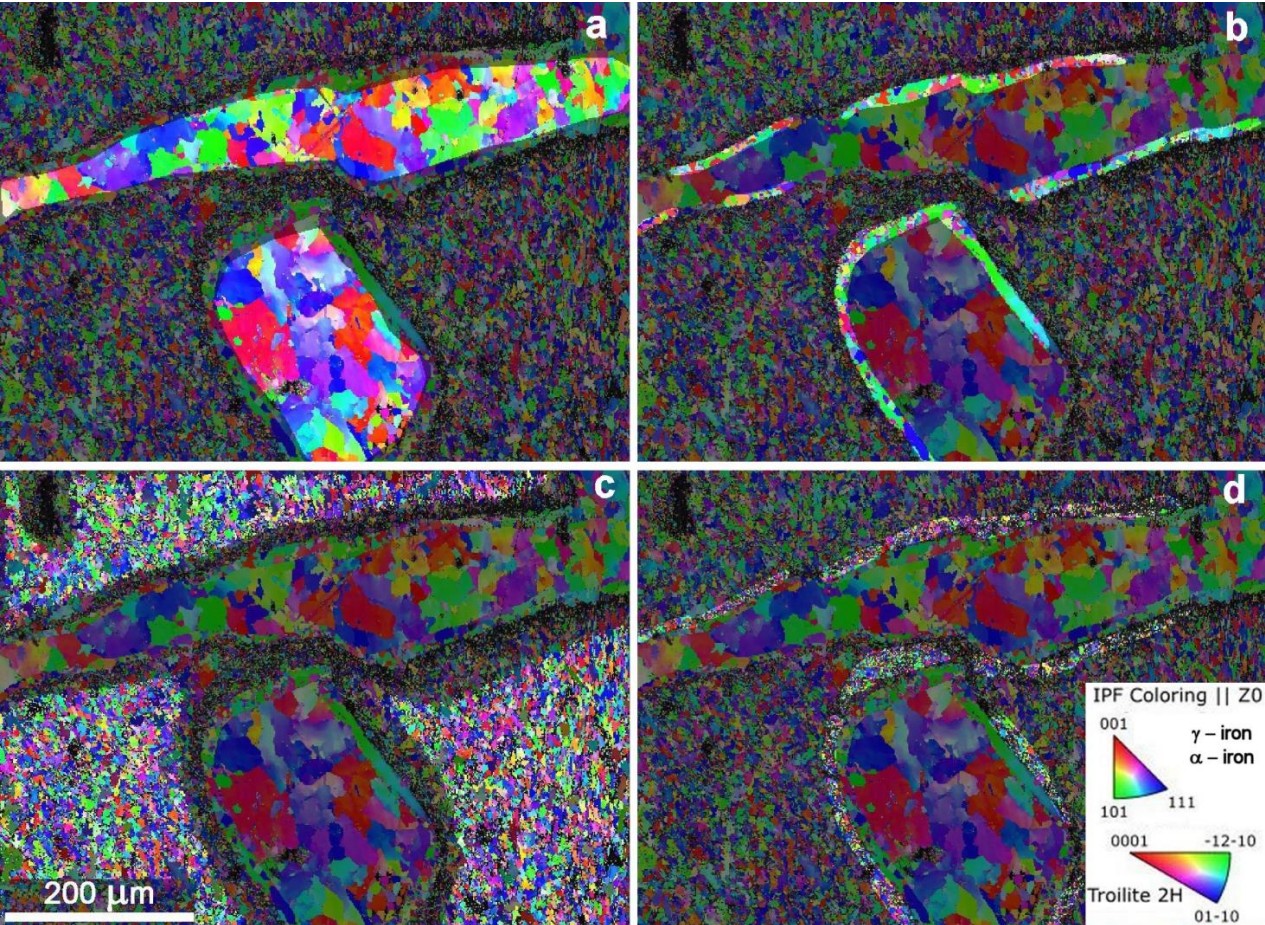

**Figure 6.** EBSD grain domains orientation maps of the area represented in Figure 5a; the colors represent the orientation of the selected crystallographic planes according to a projection parallel to the Z-axis for both the alpha and gamma phases; (**a**) coarse-grained alpha phase-rich area; (**b**) medium-grained, gamma-phase-rich area; (**c**) fine-grained plessitic area; (**d**) extra-fine-grained plessitic area.

Band contrast (BC) images (Figure 7), which resemble SEM-BSE images, were also used to check the quality of the EBSD pattern and to show the areal distribution of bcc, fcc and other phases. Regarding the phase distribution, Figure 7 shows the combined band contrast and the BSE images of two areas of the polished Oglat Sidi Ali sample at different scales. The surfaces of both areas consisted of an aggregate of Fe-Ni phases (either bcc-alpha phase or fcc-gamma phase) with minor small troilite and nickel-phosphide grains. The large homogeneous areas of the alpha phase lamellae appeared to be surrounded by rims of the gamma phase, while the remaining areas consisted of a fine-grained intermixed

aggregate of bcc-alpha and fcc-gamma phases, which are commonly defined as a plessitic texture [12], with the alpha phase prevailing over the gamma one.

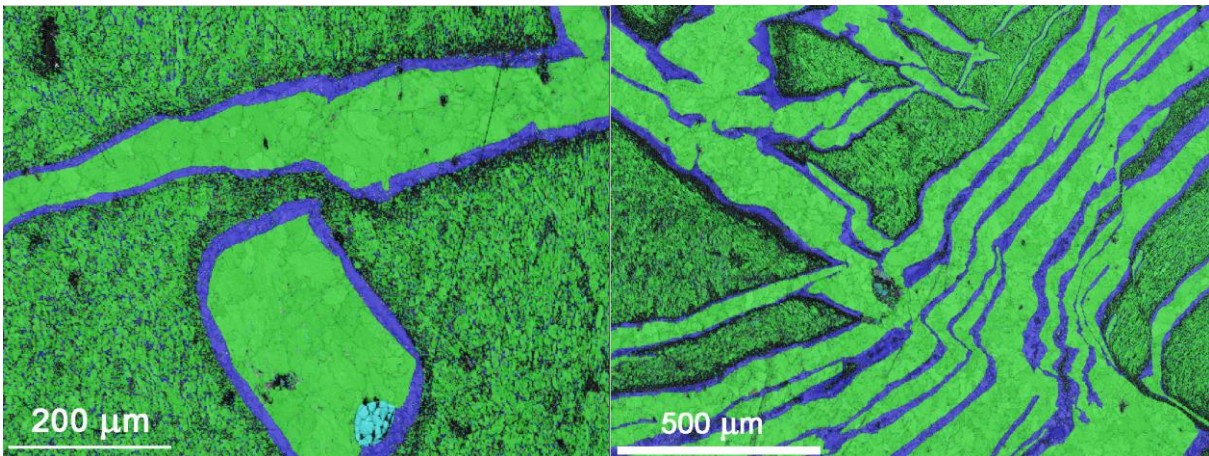

**Figure 7.** Band contrast and phase maps of two selected areas of a polished sample of the Oglat Sidi Ali meteorite; green is the alpha phase (kamacite); blue is the gamma phase (taenite or Ni-rich taenite); pale blue is the troilite (**left**) and nickel-phosphide (**right**).

Regarding the grain size, four distinct dimensional clusters were observed: coarse-grained (CG), medium-grained (MG), fine-grained (FG) and extra-fine-grained (EFG) areas (Figure 6), which corresponded to the bcc, fcc and plessitic (both FG and EFG) phases, respectively. Table 2 reports the mean grain size of these different areas. The marked difference in grain size is in good agreement with literature data [12] and can be explained by the occurrence of a process of slow cooling of a pristine mixed liquid phase with the formation of coarse-grained alpha phase, and the subsequent exsolution of the surrounding gamma phase.

**Table 2.** Grain size distribution calculated for the four areas of the Oglat Sidi Ali meteorite as distinguished in Figure 6.

| Data Set (Figure 6) | Grain Size | Main Phase | Avg. Area ($\mu m^2$) | Max. Area ($\mu m^2$) | Grain Counts |
|---|---|---|---|---|---|
| 8a | Coarse | Kamacite | 35.6 | 1987.4 | 1153 |
| 8b | Medium | Taenite | 8.5 | 179.1 | 1175 |
| 8c | Fine | Plessitic texture | 6.5 | 120.0 | 19819 |
| 8d | Extrafine | Plessitic texture | 2.9 | 37.2 | 3514 |

In contrast, the microtexture of the plessitic matrix would be the consequence of the fast cooling of martensite ($\alpha 2$) plates in a residual fcc taenite matrix at a microscale [12,13], which is in good agreement with the Fe-Ni alloys phase diagram [14]. As seen in Figure 7 (right), the alpha phase lamellae showed an evident distortion from the original parallel alignment, which suggested a post-formational shearing process with consequent deformation that was confirmed via the analysis of the elemental distribution shown in the X-ray maps (Figure 8). A clear Ni-enrichment along the kamacite–taenite borders appeared in both areas, which is in good agreement with the exsolution process of the formation of kamacite lamellae reported in the literature [12,15].

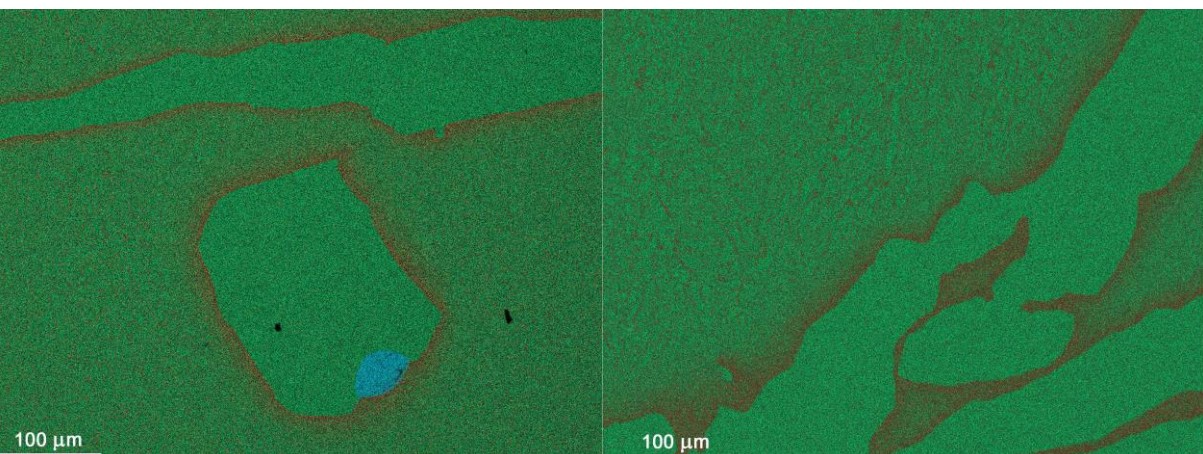

**Figure 8.** X-ray maps for Fe (green), Ni (red) and S (blue) of two different areas of the Oglat Sidi Ali meteorite; a marked Ni enrichment is visible at the kamacite–taenite border.

A detailed analysis of the orientation of the crystallites in the areas shown in Figure 6 enabled a hypothesis for a multistep formation process for the meteorite. Despite an expected iso-orientation of both the bcc alpha phase lamellae and the surrounding fcc gamma phase, according to the cooling behavior that is typical of other iron meteorites [12], a marked misorientation can be observed in both phases at any grain size. As shown in Figure 6, although most of the grains displayed a blue color, indicating a predominance of an orientation along the (111) axis, other orientations were observed in both the alpha phase and in the gamma and plessitic phases. Moreover, a clear structural distortion of the larger grains was visible in the alpha phase that displayed a fainting orientation color. This textural feature can be interpreted as due to a shock process that misoriented the previous preferred (111) orientation, similar to that reported by Yang et al. [14] for the Hammond meteorite.

These data were confirmed by the analysis of the inverse pole maps (Figure 9) of the areas in Figure 6. In particular, the (110) polar maps of the bcc-alpha phase of the coarse-grained area and the (111) polar maps of the fcc-gamma phase of the medium-grained area showed a possible original common orientation, which suggested the existence of previously established orientation relationships, such as the Kurdjumov–Sachs pattern [12]. However, they also displayed slighter discrepancies, which suggested alternative grain orientations that could be interpreted as being due to post-formational shock episodes. Furthermore, the (111) maps of the plessitic area showed no clear correspondence with the (111) maps of the coarse-grained area, which was probably due to the fast cooling experienced by the former phase compared with the other two, which did not allow for a clear ordering of the domains.

### 5.3. Mineral Chemistry

Replicate SEM-EDX spot analyses performed on the bcc-alpha (kamacite), fcc-gamma (taenite) and plessitic phases of the Oglat Sidi Ali meteorite confirmed the existence of three main compositional clusters for the CG, MG and FG areas that corresponded to the three most common mineralogical phases in these areas. Furthermore, these analyses revealed the presence of Ni-rich taenite grains inside the taenite-rich area that were mainly distributed along the taenite–kamacite boundaries, which confirmed the exsolution process evidenced by X-ray maps. The raster analyses on wide (500 × 400 μm) areas enabled the estimation of a bulk composition that was very similar to that of the plessitic, fine-grained phase. Table 3 reports the analytical data for the kamacite, taenite and plessitic-texture phases, together with other accessory phases.

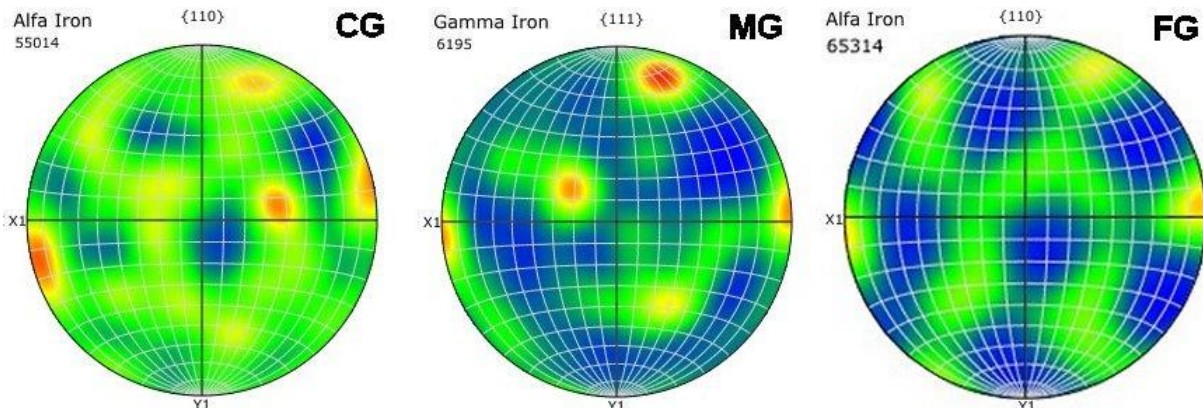

**Figure 9.** Polar maps of the area represented in Figure 6a comparing the (110) orientation of the alpha phase in the coarse and fine-grained areas with the (111) orientation of the gamma phase in the medium-grained area. A distorted K-S pattern can be observed in the first and second maps.

**Table 3.** EMPA-WDS and SEM-EDX data on the major and minor elements for raster areas and selected phases of the Oglat Sidi Ali meteorite; all data are in wt.%; (—) denotes below detection limit. Detection limits for minor elements in EMPA-WDS analyses are the following (all in wt.%): Ge = 0.049, Cu = 0.011, Si = 0.003, P = 0.003, Cr = 0.007 and S = 0.006; the D.L. for Ge in SEM-EDX raster analyses is 0.05 wt.%.

| Elements | Raster Analyses on 500 × 400 µm-Wide Areas (SEM-EDX) | Kamacite | Taenite | Ni-Rich Taenite | Plessitic Texture | Plessitic Texture | Troilite | Nickel-Phosphide |
|---|---|---|---|---|---|---|---|---|
| N° of analyses | 9 | 20 | 21 | 9 | 6 | 20 | 3 | 3 |
| Selected Area | - | Coarse-grained | Medium-grained | Medium-grained | Extrafine-grained | Fine-grained | - | - |
| Fe | 81.98 | 90.47 | 72.13 | 58.49 | 80.19 | 82.34 | 61.17 | 34.93 |
| Co | 1.72 | 1.61 | 1.26 | 0.62 | 1.42 | 1.39 | 0.47 | 0.43 |
| Ni | 15.98 | 6.17 | 24.49 | 39.02 | 16.05 | 14.40 | 0.41 | 48.12 |
| Ge | 0.25 | 0.18 | 0.24 | 0.39 | 0.13 | 0.17 | — | — |
| Cu | — | — | 0.06 | 0.10 | 0.03 | 0.03 | — | — |
| Si | — | — | — | — | — | — | — | 0.11 |
| P | — | — | — | — | — | — | 0.13 | 16.40 |
| S | — | — | — | — | — | — | 37.72 | — |
| Cr | — | — | — | — | — | — | 0.12 | — |
| Total | 99.93 | 98.45 | 98.19 | 98.64 | 97.83 | 98.34 | 100.02 | 99.99 |

The taenite featured a mean Ni content of 24.4 wt.%, with a maximum of 39.0 wt.% for Ni-rich taenite grains; the kamacite showed a mean Ni content of 6.2 wt.%; and the plessitic area displayed a mean Ni content of 14.4 wt.%, although marked variations were detected in this area due to the fine intergrowth of taenite and kamacite grains.

The bulk analyses of the major, minor and traces elements, which are requested for classifying the iron meteorites [16] resulted in high Ni (163.4) and Co (12.2) contents, both in mg/g, as well as high Cu (325.7), Ga (79.2), Ru (52.81), Pd (8.87), Au (5.21), Pt (45.30) and As (45.4) contents and relatively low W (6.46), Ir (3.33), Re (0.34) and Os (0.25) contents, all in µg/g (Table 4). The Ge concentration could not be determined using ICP-MS with a high degree of certainty. However, an estimate of the overall Ge content (2256 µg/g) was obtained by means of PIXE analysis.

Twelve groups of iron meteorites are currently recognized and designated by roman numerals (I, II, III, IV) and letters A through F according to the concentrations of selected siderophile trace elements (such as Ga and Ir) plotted against the overall Ni content on a logarithmic plot [17]. Most plessitic octahedrites belong to either the IIC or IIF chemical groups [16]. Although the texture of the Oglat Sidi Ali meteorite showed characteristics

similar to those of other plessitic octahedrites, the high Ni, Ga, Ru, Pd and Pt contents, as well as the relatively low Ir contents, were outside the limits for the IIC or IIF groups (Figure 10). For this reason, we confirmed the classification as an ungrouped plessitic octahedrite iron meteorite [18].

**Table 4.** Siderophile elements composition of the Oglat Sidi Ali, NWA 859, NWA 11010, NWA 7335 and Butler meteorites (data from Goldstein [21], Buchwald [22], Wasson [20], Moggi et al. [18], Bouvier et al. [19] and Ruzicka et al. [23]). * Estimated value from PIXE analysis.

| Elements Meteorites | Ni (mg/g) | Co (mg/g) | Ge (µg/g) | Cu (µg/g) | Ga (µg/g) | As (µg/g) | W (µg/g) | Ir (µg/g) | Pt (µg/g) | Au (µg/g) |
|---|---|---|---|---|---|---|---|---|---|---|
| Oglat Sidi Ali | 163.4 | 12.2 | 2256 * | 325.7 | 79.2 | 45.4 | 6.7 | 3.3 | 45.3 | 5.2 |
| NWA 859 | 159.3 | 13.1 | 2200 | 296.1 | 87.0 | 54.2 | 6.8 | 2.5 | 37.7 | 6.5 |
| NWA 11010 | 160.7 | 12.8 | 2666 | 245.2 | 87.5 | 53.3 | 7.0 | 2.5 | 41.0 | 6.5 |
| Butler | 157.2 | 10.3 | 2000 | 151.1 | 87.1 | 48.2 | 5.1 | 1.8 | 34.9 | 6.8 |
| NWA 7335 | 109.8 | 8.4 | 41.2 | 229 | 50.8 | 8.7 | 3.7 | 12.3 | 26.3 | 1.7 |

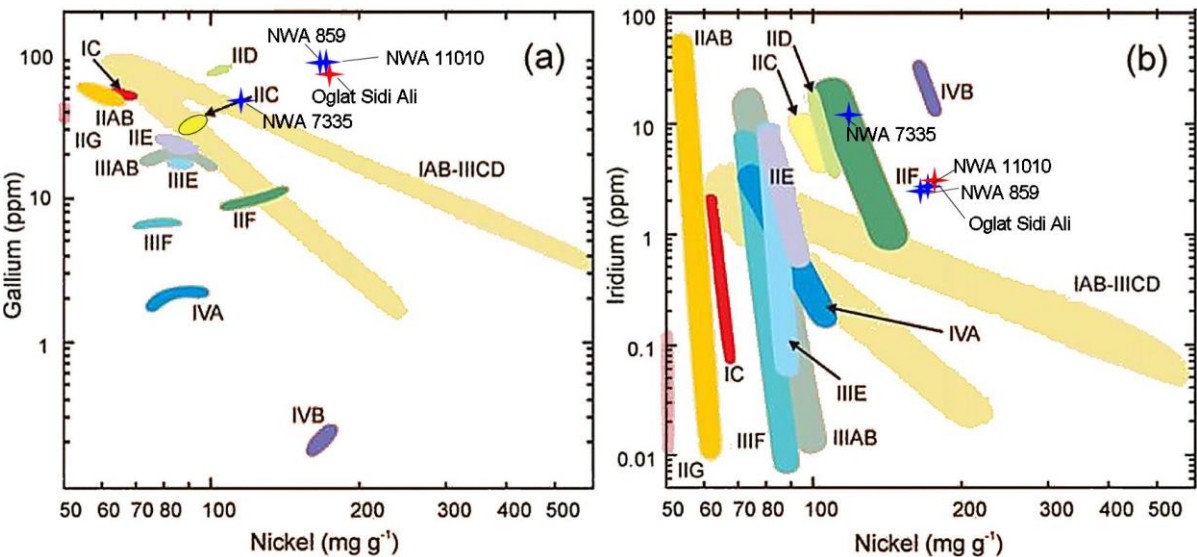

**Figure 10.** Logarithmic plot of the bulk amounts of Ga (**a**) and Ir (µg/g) (**b**) versus Ni (mg/g) of the Oglat Sidi Ali meteorite and some NWA ungrouped irons and literature data for various iron meteorite groups (modified based on Grady et al. [16]).

Plotting the Ga and Ir contents of the Oglat Sidi Ali meteorite and those of other ungrouped iron meteorites recovered in the same area, namely, NWA 859 (also known as Taza), NWA 11010 [19] and NWA 7335, resulted in strong geochemical affinities between the Oglat Sidi Ali, NWA 859 and NWA 11010 [19] meteorites, whereas differences were evidenced for NWA 7335 (Figure 10).

Furthermore, a comparison of the literature data for a set of siderophile minor and trace elements in these meteorites (Table 4) showed that the contents of the elements were similar, thus suggesting a possible common genetic origin for all these ungrouped plessitic iron meteorites, as previously proposed by Wasson [20]. In particular, all these iron meteorites featured relatively high values of Ni (141–161 mg/g), Ge (2000–2666 µg/g), Ga (79–87 µg/g) and Pt (35–44 µg/g) and relatively low Ir (1.8–2.5 µg/g) contents, as well as plessitic octahedrite structures.

In order to test a possible pairing between these meteorites, the compositional data for eight main siderophile elements (Ni, Co, Ge, Cu, Ga, W, Ir and Pt) were compared with those of the ungrouped iron Butler meteorite (Figure 11). NWA 859 and NWA 11010 have

compositions that are quite similar to that of Oglat Sidi Ali, while NWA 7335 has marked differences in Ni, Co, Ga, W and Pt contents. Regarding the textural features, the Oglat Sidi Ali meteorite showed a plessitic structure similar to that of NWA 859 and NWA 11010 at both the hand specimen scale and at the microscopic scale (Figure 12), with spindles of kamacite separated by plessitic fields similar to those already described by Buchwald [22] and Wasson [20] for the Butler ungrouped iron meteorite. Thus, based on the presently available chemical and textural evidence, it seems probable that NWA 859 and NWA 11010 were paired with Oglat Sidi Ali, while NWA 7335 has to be considered a separate find. Oglat Sidi Ali, NWA 859 and NWA 11010 may be genetically related to Butler.

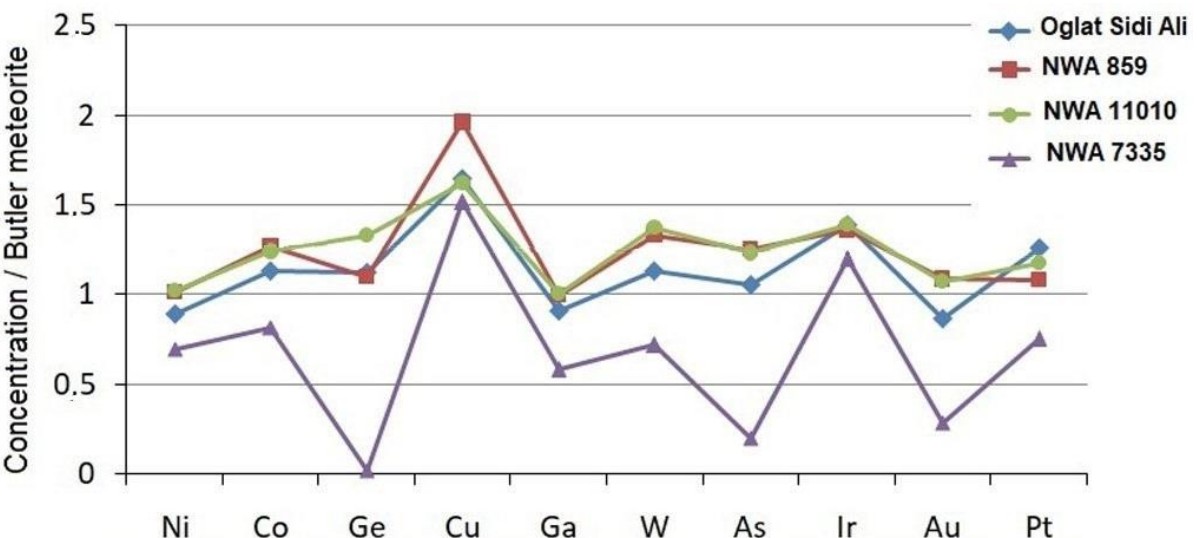

**Figure 11.** Ratios of the siderophile element contents of selected NWA ungrouped iron meteorites normalized to those of the Butler meteorite [21].

Although many other ungrouped iron meteorites are found in the NWA group (NWA 2428, NWA 4702, NWA 4705, NWA 6163, NWA 6166, NWA 6167, NWA 6583, NWA 6932, NWA 7335), some of which show plessitic textural features, their chemical compositions [20,24] are quite different and they do not appear genetically related to the Oglat Sidi Ali meteorite. Moreover, the lack of data concerning the exact finding locations does not allow any further consideration concerning possible pairings [25].

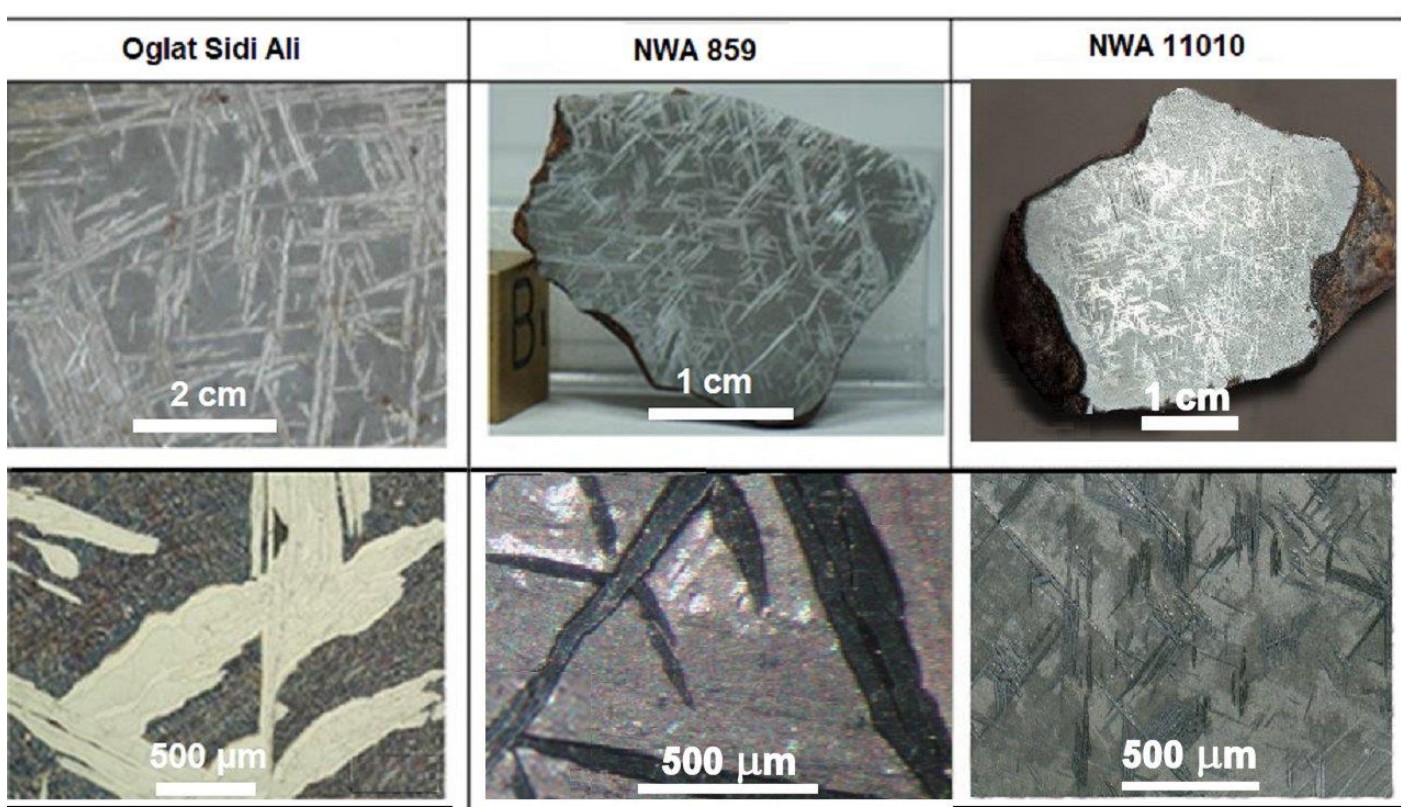

**Figure 12.** Hand-size specimen and stereomicrograph images of the textures of the Oglat Sidi Ali, NWA 859 and NWA 11010 meteorites. NWA 11010 photos courtesy of T. DeLisle, Pisgah Astronomical Research Institute (hand-size specimen) and by A. Love, Appalachian State University, sample# ASU-NWA-98 (stereomicrograph image).

## 6. Conclusions

The distribution of meteorites found at the Oglat Sidi Ali site, as well as their smoothed, regmaglypted characteristics, suggested that they represent a strewnfield formed by a single fragmentation event of a single body that broke up in the lower atmosphere. There was no evidence for "shrapnel-like" fragments resulting from the impact of a large object on the surface, as observed in other cases (e.g., Gebel Kamil) [26–28]. Although circular depressions or Dayas are very common in the meteorite strewnfield area, these cannot be considered impact structures based on the lack of evidence of impact metamorphism (shatter cones, shocked quartz, etc.) and the lack of "shrapnel-like" fragments. The circular depressions in the area are very likely to be dolines, thus only further field research can provide more detailed information about the identification of the karst landscapes, describe karst features and detect geological structures relevant to karst development.

Computer simulations [10] of the strewnfield formation show that the scattering area length should be greater than 11 km at an entry angle lower than 45°. Based on the presence of only individual fragments in the Oglat Sidi Ali strewnfield, the speed of the meteoroid should have been low, resulting in a large amount of relatively small fragments. Based on the mass distribution of meteorites, the flight path of the meteoroid was from NE to SW. The total estimated collected mass of meteorite fragments exceeded 800 kg.

From a genetic point of view, the crystallographic orientation data obtained from EBSD analyses, together with the microtextural appearance, suggested a possible two-stage formation according to an exsolution model that is common to that of many other iron meteorites, followed by a shock process that was possibly due to the ejection impact that partly modified the original textural features. Finally, the freshness of the fragments found suggested that the age of the Oglat Sidi Ali meteorite shower was relatively recent [29], i.e., it occurred in the Quaternary age. Although the textural data of the Oglat Sidi Ali meteorite

showed similarities to those of other plessitic octahedrites, geochemical data suggested it did not belong to the IIC group but joined an increasing number of ungrouped plessitic octahedrites that include NWA 859 (Taza) and NWA 11010, which were recovered in the same geographical area and suggesting possible pairing among these and Butler, which may originate from the same parent body.

**Author Contributions:** Conceptualization, H.N., A.I, V.M.C., G.P. and G.S.S.; methodology, H.N., A.I., V.M.C., G.P., C.D.K.H. and G.S.S.; formal analysis, A.I., V.M.C., G.P. and C.D.K.H.; investigation, H.N., A.I., M.E.-n., V.M.C., G.P., C.D.K.H. and G.S.S.; data curation, H.N., A.I., V.M.C., G.P., C.D.K.H. and G.S.S.; writing—original draft preparation, H.N., A.I., V.M.C., G.P., C.D.K.H. and G.S.S.; writing—review and editing, H.N., A.I., M.E.-n., V.M.C., G.P., C.D.K.H. and G.S.S.; supervision, H.N., A.I., V.M.C., G.P. and G.S.S. All authors have read and agreed to the published version of the manuscript.

**Funding:** This research received no external funding.

**Data Availability Statement:** Not applicable.

**Acknowledgments:** We thank the Moussaoui family, Bassou Moulay Ali, Ainouch Mohamed and the inhabitants of the commune of Maatarka for their hospitality and their assistance in the collection of information. We also thank Massimo Chiari for providing PIXE data on the Ge contents and Tim DeLisle (Pisgah Astronomical Research Institute) and Anthony Love (Geological and Environmental Sciences, Appalachian State University) for providing images of NWA 11010. Special thanks are due to Mario Di Martino for his first-sight confirmation of the meteoritic origin of the type specimen samples. Minor and trace element analyses at the University of Alberta were funded by the Natural Sciences and Engineering Research Council of Canada Grant RGPIN-2018-04902 to C.D.K.H.

**Conflicts of Interest:** The authors declare no conflict of interest.

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
