# Peer review of "Minerochemical and Microtextural Study of the Ungrouped Iron Meteorite Oglat Sidi Ali, Eastern Highlands, Morocco, and Geomorphological Characterization of Its Strewnfield"

_minerals, doi:10.3390/min12111470_

Round 1

Reviewer 1 Report

The paper reports the physical characteristics, textural and chemical features of the ungrouped iron meteorite Oglat Sidi Ali with comparison with other ungrouped iron meteorites found in various cities of North-East Morocco. Research based on numerous meteorite samples discovered during three fieldwork expeditions. Authors also estimate aerodynamics of the meteoroid fall, its size and age of the Oglat Sidi Ali meteorite shower. The paper is well written and designed. However, several points should be addressed before publication:

1.      Authors describes many founds of Oglat Sidi Ali samples, but there is no information about the quantity of samples analyzed by SEM-EDS, EBSD, ICP-MS and PIXE techniques.

2.       Authors conclude that the Oglat Sidi Ali meteorite shower occurred in the Quaternary age because of freshness of the meteorite fragments, but it remains unclear are there secondary minerals at all? What is the weathering grade of the fragments (Wlotzka, 1993)?

3.      The same for the shock stage of the Oglat Sidi Ali and other mentioned ungrouped iron meteorites.

4.      Also authors can improve this paper by adding (in text and on a map) the distance from Oglat Sidi Ali strewnfield to other meteorites found sites.

Author Response

Reviewer 1

The paper reports the physical characteristics, textural and chemical features of the ungrouped iron meteorite Oglat Sidi Ali with comparison with other ungrouped iron meteorites found in various cities of North-East Morocco. Research based on numerous meteorite samples discovered during three fieldwork expeditions. Authors also estimate aerodynamics of the meteoroid fall, its size and age of the Oglat Sidi Ali meteorite shower. The paper is well written and designed.

However, several points should be addressed before publication:

Point 1: Authors describes many founds of Oglat Sidi Ali samples, but there is no information about the quantity of samples analyzed by SEM-EDS, EBSD, ICP-MS and PIXE techniques.

Response 1: 1 specimen (MSN-Fi I3665) for SEM-EDS, EBSD, and PIXE analysis; 1 specimen (another chip from I3665) for ICP-MS.

Point 2: Authors conclude that the Oglat Sidi Ali meteorite shower occurred in the Quaternary age because of freshness of the meteorite fragments, but it remains unclear are there secondary minerals at all? What is the weathering grade of the fragments (Wlotzka, 1993)?

Response 2: The weathering scale of Wlotzka (1993) regards Ordinary Chondrites, and its application to iron meteorites is doubtful. However, iron oxides amount at about 20% surface area. Although the surfaces of many specimens are reddish, the cut surfaces of interiors of most samples are fresh and display no traces of staining (no secondary minerals, new Fig. 6), suggesting alteration grade W1 according to the classification of Wlotzka (1993) (lines 209 and 210).

Point 3: The same for the shock stage of the Oglat Sidi Ali and other mentioned ungrouped iron meteorites.

Response 3: Difficult to say for an iron meteorite, and estimates are made on the basis of external features and lamellae deformation in new Fig. 6. The shock scale is projected for silicates, and we suppose this is a weak shock based on the conservation of the initial texture of the meteorite, according to experimental studies by Eiji Ohtani et al., (2022); (Cf. reference below).

Ohtani, E., Sakurabayashi, T. & Kurosawa, K. Experimental simulations of shock textures in BCC iron: implications for iron meteorites. Prog Earth Planet Sci 9, 24 (2022).

Point 4: Also authors can improve this paper by adding (in text and on a map) the distance from Oglat Sidi Ali strewnfield to other meteorites found sites.

Response 4: The found sites of ungrouped plessitic octahedrites NWA 859 (Taza) and NWA 11010 are unknown, and they have been sold in different cities of Morocco (Oujda, Taza, Sahara, etc.). This is the reason why they are named NWA.

Reviewer 2 Report

There is no doubt that desert meteorites play an increasingly important role in the study of cosmochemistry due to their relatively low recovery costs and generally unrestricted collection and flow in most parts of the world. Several countries have collected a large number of meteorites in the Antarctic. However, most of the Antarctic meteorites have changed their positions after they landed on the earth’s surface, so it is almost impossible to obtain information such as the entry angle, entry direction, and breakup of meteoroids. In most cases, the locations of desert meteorites represent the landing sites, which are important information for studying the mechanism of meteoroid entry and break up. Meanwhile, the location information is significantly important to judge whether the meteorites are pairing or independent falls. 

The authors of this paper conducted a mineralogical and geochemical study of a sample of the Moroccan Eastern Highlands Oglat Sidi Ali iron meteorite Strewnfied. The authors believe that the meteoroid of the Oglat Sidi Ali meteorite descended from NE-SW and broke up at a lower altitude. They also suggested that the iron meteorites collected in Oglat Sidi Ali strewnfield are paired with NWA 859 and NWA 11010. The paper is well-organized and also in the scope of the journal of Minerals. I recommend this paper for publication. 

Here are some minor suggestions:

1)    Line 20-21: Delete the sentence “Enigmatic circular …Unknown.”.

2)    Line 130-131: “An entry angle of 90°produces a circular scattering area, whereas lower entry angles 130 determine elongated elliptical strewn fields.”

This need to be clarified. Is this based on assuming or any model? 

3)    Figure 5: The scar bar for each meteorite should be added;

4)    Figure 6: The scale bars should be fixed at the same position in each panel;

5)    Figure 7 and Figure 8: A space is necessary between any two pictures;

6)    Figure 14: These images need to be well prepared. Tow scale bars in the middle of the bottom panel.

7)    Line 414-415: If these ungrouped NWA irons were collected in different locations, together with Butler, they could establish a grouplet. 

Author Response

Reviewer 2

There is no doubt that desert meteorites play an increasingly important role in the study of cosmochemistry due to their relatively low recovery costs and generally unrestricted collection and flow in most parts of the world. Several countries have collected a large number of meteorites in the Antarctic. However, most of the Antarctic meteorites have changed their positions after they landed on the earth’s surface, so it is almost impossible to obtain information such as the entry angle, entry direction, and breakup of meteoroids. In most cases, the locations of desert meteorites represent the landing sites, which are important information for studying the mechanism of meteoroid entry and break up. Meanwhile, the location information is significantly important to judge whether the meteorites are pairing or independent falls. 

The authors of this paper conducted a mineralogical and geochemical study of a sample of the Moroccan Eastern Highlands Oglat Sidi Ali iron meteorite Strewnfied. The authors believe that the meteoroid of the Oglat Sidi Ali meteorite descended from NE-SW and broke up at a lower altitude. They also suggested that the iron meteorites collected in Oglat Sidi Ali strewnfield are paired with NWA 859 and NWA 11010. The paper is well-organized and also in the scope of the journal of Minerals. I recommend this paper for publication. 

Here are some minor suggestions:

Point 1: Line 20-21: Delete the sentence “Enigmatic circular …Unknown.”.

Response 1: The sentence has been deleted.

Point 2: Line 130-131: “An entry angle of 90°produces a circular scattering area, whereas lower entry angles 130 determine elongated elliptical strewn fields.”

This need to be clarified. Is this based on assuming or any model? 

Response 2: The assessment is based on the model proposed by Lorenz et al. (2015) (Cf. reference below).

Lorenz, C.A., Ivanova, M.A., Artemieva, N.A., Sadilenko, D.A., Chennaoui, A.H., Roschina, I.A., Korochantsev, A.V. and Humayun, M. (2015) Formation of a small impact structure discovered within the Agoudal meteorite strewn field, Morocco. Meteoritics & Planetary Science, 50, 112–134. 

Point 3: Figure 5: The scar bar for each meteorite should be added;

Response 3: The scar bar has been added.

Point 4: Figure 6: The scale bars should be fixed at the same position in each panel;

Response 4: Done.                                                                        

Point 5: Figure 7 and Figure 8: A space is necessary between any two pictures;

Response 5: Done.

Point 6: Figure 14: These images need to be well prepared. Tow scale bars in the middle of the bottom panel.

Response 6: Done.

Point 7: Line 414-415: If these ungrouped NWA irons were collected at different locations, with Butler, they could establish a grouplet.

Response 7: Oglat Sidi Ali, NWA 859 (also known as 366 Taza) and NWA 11010 would not form a grouplet because apparently they have similar geochemical data and are also supposed to have been recovered in the same location, and would result to be paired meteorites.

Reviewer 3 Report

Minerochemical and microtextural study of the ungrouped iron meteorite Oglat Sidi Ali, Eastern Highlands, Morocco, and geomorphological characterization of its strewnfield

Hassane Nachit, Abderrahman Ibhi, Mohamed En-nasiry, Vanni Moggi Cecchi, Giovanni Pratesi, Christopher  D.K. Herd and Giorgio S. Senesi,

Work centered on studying different fragments of an iron  meteorite is presented. The authors focus mainly on defining the dimensions and geomorphology of the meteorite strewnfield, studying the mineralogy, geochemistry, and microtexture of the same meteorite fragments to finally classify the meteorite fragments establishing their genetic relationship with other previously studied meteoritic fragments. The topic is interesting and provides data from different scales of work using classic characterization tools (optical microscopy) and the latest generation (EBSD).

The manuscript is well organized, but given the relevance of the material studied, some aspects related to the format of the images and tables mentioned below should be refined:

Figure 1. Indicate the provenance of the geological map. It is worth mentioning that, given the study's objectives, it would be interesting to incorporate the main geomorphological features of the area into the maps.

Figure 2. The so-called "Daya" are not very well observed; it would be good to indicate the layout of these in the photograph. Image 2b is not scaled.

Lines 95-137. The information entered in this paragraph should also be summarized as a table.

Lines 142-149: The authors should briefly explain why some important pieces were deposited in Italian museums or collections. It is worth mentioning that many countries have very strict regulations that allow the pieces always to be preserved in national collections.

Line 150: indicate the model of the Zeiss microscope used in petrographic observation.

Figure 5: The image is very poor and must be reconstructed, indicating the scale of the pieces. At the same time, it is recommended to include as an annex a small catalog (dimensions, weight, description, and photograph) of the 72 fragments found in the field campaigns that the authors report having found.

In Figure 6, details mentioned in the text should be indicated by abbreviations. Unfortunately, the figure is poorly made.

Figure 8 has a legend related to the orientation of the crystals in the upper-left corner that cannot be read well.

Line 279:  include the corresponding references.

Author Response

Reviewer 3

Minerochemical and microtextural study of the ungrouped iron meteorite Oglat Sidi Ali, Eastern Highlands, Morocco, and geomorphological characterization of its strewnfield

Hassane Nachit, Abderrahman Ibhi, Mohamed En-nasiry, Vanni Moggi Cecchi, Giovanni Pratesi, Christopher  D.K. Herd and Giorgio S. Senesi,

Work centered on studying different fragments of an iron  meteorite is presented. The authors focus mainly on defining the dimensions and geomorphology of the meteorite strewnfield, studying the mineralogy, geochemistry, and microtexture of the same meteorite fragments to finally classify the meteorite fragments establishing their genetic relationship with other previously studied meteoritic fragments. The topic is interesting and provides data from different scales of work using classic characterization tools (optical microscopy) and the latest generation (EBSD).

The manuscript is well organized, but given the relevance of the material studied, some aspects related to the format of the images and tables mentioned below should be refined:

Point 1: Figure 1. Indicate the provenance of the geological map. It is worth mentioning that, given the study's objectives, it would be interesting to incorporate the main geomorphological features of the area into the maps.

Response 1: The strewnfield in the Eastern Highlands of Morocco indicated in Figure 1, left, is taken after the Geological map of High Plateaus after the geological map of Morocco at 1/1,000,000. The circular structures called Dayas and the dispersed field of meteorite fragments (right) taken from google maps. The locality of Oglat Sidi Ali is located in a vast flat plateau without notable relief, i.e., it has a flat topography.

Point 2: Figure 2. The so-called "Daya" are not very well observed; it would be good to indicate the layout of these in the photograph. Image 2b is not scaled.

Response 2: Figure 2 has been deleted.

Point 3: Lines 95-137. The information entered in this paragraph should also be summarized as a table.

Response 3: As requested by the Reviewer, new Table 1, which summarizes the information in Section 3, lines 95-137, has been included in the main text.

Point 4: Lines 142-149: The authors should briefly explain why some important pieces were deposited in Italian museums or collections. It is worth mentioning that many countries have very strict regulations that allow the pieces always to be preserved in national collections.

Response 4: The main mass is preserved in a Moroccan national collection, namely in the Laboratory of Petrology, Mineralogy and Materials, Ibn Zohr University, Faculty of Sciences, Agadir, BP 8106, Morocco. The Museum of Natural History of the University of Firenze (Italy) only preserves the type specimen because it has to be deposited where it has been studied.

Point 5: Line 150: indicate the model of the Zeiss microscope used in petrographic observation.

Response 5: A first hand textural investigation has been performed with a Zeiss metallographic optical microscope (Axio Imager.D2) at the Laboratories of the Ibn-Zohr University, and then by an Axioscope 5 in Italy.

Point 6: Figure 5: The image is very poor and must be reconstructed, indicating the scale of the pieces. At the same time, it is recommended to include as an annex a small catalog (dimensions, weight, description, and photograph) of the 72 fragments found in the field campaigns that the authors report having found.

Response 6: We believe that the images are very good. The scale of the pieces has been included. Some information of the 72 fragments found in the field campaign is reported in the newly included Table 1.

Point 7: In Figure 6, details mentioned in the text should be indicated by abbreviations. Unfortunately, the figure is poorly made.

Response 7: Done.

Point 8: Figure 8 has a legend related to the orientation of the crystals in the upper-left corner that cannot be read well.

Response 8: The legend in the upper left corner of Figure 8 has been enlarged.

Point 9: Line 279:  include the corresponding references.

Response 9: Done.

Reviewer 4 Report

This manuscript focuses on an Oglat Sidi Ali iron meteorite. Through the study of its falling point, mineral structures and chemical composition, the author gives five conclusions:

 (1) The Oglat Sidi Ali iron meteorite is one fragment of a unique meteorite strewnfield originated from an iron meteorite shower by fragmentation of a single body that broke up in the lower atmosphere; (2)The Oglat Sidi Ali iron meteorite fragments spread across the NE–SW oriented, 20-km-long and 5-km-wide strewnfield; (3) Enigmatic circular depressions occur in the strewnfield area with an origin still unknown; (4) Geochemical and mineralogical data achieved on Oglat Sidi Ali fragments, as well as the analysis of its microstructures obtained by electron backscattered diffraction (EBSD) suggest it is classified as an ungrouped iron meteorite; (5) The mineralogy, geochemistry and textural features of the Oglat Sidi Ali iron meteorite is similar as those of NWA 859 and NWA 11010 iron meteorites, which suggest a common origin from a single extraterrestrial body.

Overall, there are a lot of things tries to be addressed in this paper. Nevertheless, there is not enough evidence to support these conclusions.

For example, as for a meteorite shower with a 20-km-long and 5-km-wide strewnfield,the authors should provide a sufficient number of iron meteorites with the same classification type, and similar chemical compositions. And also these iron meteorites should be distributed in different point of the strewfield. The accurate information of qualified iron meteorites should be provided, such as: name, number, size, weight, longitude and latitude. Obviously, only three chemically similar iron meteorites (Oglat Sidi Ali, NWA 859 and NWA 11010) do not prove the authors' conclusions.

It is suggested that the authors divide the content of this paper into three papers: (1) on chemical classification; (2) About meteorite showers; And (3) about enigmatic circular depressions.

Specific comments:

Figure 1., the “dayas” called by author have no obviously orientation (NE-SW) distribution trend and morphology. They look more like they're evenly distributed over the area of the graph.

Figure 2 is not clear enough to make out the topographic features with “dayas”.

Page4, 2-3 paragraph: if these described meteorites are not of the same type with similar chemical composition, they cannot be used as evidence of a same meteorite shower.

Figure 4 (left) should provide the detail information of these meteorites in the strewfield, such as meteoritic name, size, classification type, and the latitude and longitude of each point.

Line 98-201 (Page 6) and line 209-211 (page 7): Repetitive descriptions should be avoided.

Figure 5 and Figure 6: some pictures are not very clear.

Line 353-354 (Page 12): authors wrote: “Ge concentration could not be determined by ICP-MS with a high degree of certainty. However, an estimate of the overall Ge content (2256 μg/g=2.25 mg/g) was obtained by means of PIXE analysis“  I am wondering why the concentration of Ge (2.256 mg/g) cannot be determined accurately. However, the higher concentration of Ni (163.4 mg/g) and Co (12.2 mg/g) can be determined accurately. According to the authors, all of these elements were determined using ICPMS. I am also wondering why such a high concentration of Ni (163.4 mg/g) could still be measured by an ICPMS, which is well known for the determination of trace elements.

Author Response

Reviewer 4

This manuscript focuses on an Oglat Sidi Ali iron meteorite. Through the study of its falling point, mineral structures and chemical composition, the author gives five conclusions:

 (1) The Oglat Sidi Ali iron meteorite is one fragment of a unique meteorite strewnfield originated from an iron meteorite shower by fragmentation of a single body that broke up in the lower atmosphere; (2)The Oglat Sidi Ali iron meteorite fragments spread across the NE–SW oriented, 20-km-long and 5-km-wide strewnfield; (3) Enigmatic circular depressions occur in the strewnfield area with an origin still unknown; (4) Geochemical and mineralogical data achieved on Oglat Sidi Ali fragments, as well as the analysis of its microstructures obtained by electron backscattered diffraction (EBSD) suggest it is classified as an ungrouped iron meteorite; (5) The mineralogy, geochemistry and textural features of the Oglat Sidi Ali iron meteorite is similar as those of NWA 859 and NWA 11010 iron meteorites, which suggest a common origin from a single extraterrestrial body.

Overall, there are a lot of things tries to be addressed in this paper. Nevertheless, there is not enough evidence to support these conclusions.

For example, as for a meteorite shower with a 20-km-long and 5-km-wide strewnfield,the authors should provide a sufficient number of iron meteorites with the same classification type, and similar chemical compositions. And also these iron meteorites should be distributed in different point of the strewfield. The accurate information of qualified iron meteorites should be provided, such as: name, number, size, weight, longitude and latitude. Obviously, only three chemically similar iron meteorites (Oglat Sidi Ali, NWA 859 and NWA 11010) do not prove the authors' conclusions.

Point 1: It is suggested that the authors divide the content of this paper into three papers: (1) on chemical classification; (2) About meteorite showers; And (3) about enigmatic circular depressions.

Response 1: We believe that the information contained in this manuscript should be kept in one paper only, and not spread into three distinct papers. Further systematic structural and geochemical studies are planned for the various meteorite fragments and also for the breccias found in the enigmatic circular depressions region to possibly reveal their origin. 

Specific comments:

Point 2: Figure 1., the “dayas” called by author have no obviously orientation (NE-SW) distribution trend and morphology. They look more like they're evenly distributed over the area of the graph.

Response 2: The related text has been modified. 

Point 3: Figure 2 is not clear enough to make out the topographic features with “dayas”.

Response 3: Figure 2 has been deleted.

Point 4: Page4, 2-3 paragraph: if these described meteorites are not of the same type with similar chemical composition, they cannot be used as evidence of a same meteorite shower.

Response 4: We confirm that our data show that all meteorite fragments found in the locality of Oglat sidi Ali have the same texture and mineralogical composition. Moreover, the analyses by SEM carried out in Italy and Morocco on different fragments are identical. Thus we can conclude that they are similar.

Point 5: Figure 4 (left) should provide the detail information of these meteorites in the strewfield, such as meteoritic name, size, classification type, and the latitude and longitude of each point.

Response 5: Figure 4 already shows the distribution of fragments of the Oglat Sidi Ali meteorite and also reports the weight of the fragments found. The new Table 1 also provides some additional information of the fragments recovered.

Point 6: Line 98-201 (Page 6) and line 209-211 (page 7): Repetitive descriptions should be avoided.

Response 6: We do not see any repetition here.

Point 7: Figure 5 and Figure 6: some pictures are not very clear.

Response 7: Apparently, these pictures are quite clear, and are the best we can provide. 

Point 8: Line 353-354 (Page 12): authors wrote: “Ge concentration could not be determined by ICP-MS with a high degree of certainty. However, an estimate of the overall Ge content (2256 μg/g=2.25 mg/g) was obtained by means of PIXE analysis“  I am wondering why the concentration of Ge (2.256 mg/g) cannot be determined accurately. However, the higher concentration of Ni (163.4 mg/g) and Co (12.2 mg/g) can be determined accurately. According to the authors, all of these elements were determined using ICPMS. I am also wondering why such a high concentration of Ni (163.4 mg/g) could still be measured by an ICPMS, which is well known for the determination of trace elements.

Response 8: The element Ge is the only one analyzed by PIXE, due to the lack of a Ge-containing standard for ICP-MS. The estimate derives from a non-standardized procedure used in PIXE analysis. All the other data are from replicate ICP-MS affordable analyses with very low standard deviation.

Round 2

Reviewer 4 Report

(1) Figure5 and 13 are unclear. 

(2) In response to the authors' conclusions “A comparison of this meteorite with 22 other ungrouped iron meteorites purchased between 2001 and 2016 in various cities of North-East 23 Morocco show apparently similar mineralogy, geochemistry and textural features to those of NWA 24 859 and NWA 11010 meteorites, which suggest a common origin from a single extraterrestrial 25 body.” There is no bulk trace element content data for other 22 meteorites, so they cannot confirm this conclusion. Similarly, for the strewfield range outlined by the author, there is not enough data to confirm that they are from a single meteorite shower.

Author Response

Reviewer 4

Point 1: Figure5 and 13 are unclear. 

Response 1: We think that the reviewer intends for “unclear” a not high resolution! As for Figure 5 we have decided to delete it. However, regarding Figure 13 we cannot improve it as they are the best images we can obtain, which have a quite good resolution for hand size specimens. In our opinion, Figure 13 is quite clear in showing what we discuss. We hope that the Editor will find it enough clear as well.

Point 2: In response to the authors' conclusions “A comparison of this meteorite with 22 other ungrouped iron meteorites purchased between 2001 and 2016 in various cities of North-East 23 Morocco show apparently similar mineralogy, geochemistry and textural features to those of NWA 24 859 and NWA 11010 meteorites, which suggest a common origin from a single extraterrestrial 25 body.” There is no bulk trace element content data for other 22 meteorites, so they cannot confirm this conclusion. Similarly, for the strewfield range outlined by the author, there is not enough data to confirm that they are from a single meteorite shower.

Response 2: Probably the reviewer is referring at the sentence present in the Abstract and not in the Conclusions. However, the sentence about the bulk trace element content data for the other meteorites is supported and justified by Table 2 present in Reference 20 (Wasson, 2011), as clearly quoted in our text, see row 411. Concerning the strewnfield range, our hypothesis is suggested by data and field observations. Anyway, following the Reviewer’s suggestion we have modified the sentence in the Conclusions (rows 422-425, blue text): “The distribution of meteorites found at the Oglat Sidi Ali site, as well as their smoothed, regmaglypted characteristics may suggest that they represent a strewn field formed by a single fragmentation event of a single body that broke up in the lower atmosphere.”

Round 3

Reviewer 4 Report

In this paper, detailed mineralogical and chemical composition analysis data of the three samples are determined, which provides a good basic data for relevant research of ungroup iron meteorites in the Oglat Sidi Ali area in the future.

Author Response

Point 1: In this paper, detailed mineralogical and chemical composition analysis data of the three samples are determined, which provides a good basic data for relevant research of ungroup iron meteorites in the Oglat Sidi Ali area in the future.

Response 1: We thank the reviewer to recognize elements of novelty in our work. Further, systematic structural and geochemical studies are planned for the various meteorite fragments and also for the breccias found in the enigmatic circular depressions region to possibly reveal their origin.